# Analysis of the emergency response capacity of nursing staff for public health emergencies and the influencing factors: A cross-sectional study in China

Xin Gu[1‡], Ping Yan[1,2], Xia Zhang[3], Hu Peng[1], Zhijie Cao[1], Lina Ma[1], Li Zhang[2,3]*

**1** School of Nursing, Xinjiang Medical University, Urumqi, Xinjiang, China, **2** Health Care Research Center for Xinjiang Regional Population, Urumqi, Xinjiang, China, **3** The First Affiliated Hospital of Xinjiang Medical University, Urumqi, Xinjiang, China

‡ Xin Gu is the first author of this article.
* 121449072@qq.com

## Abstract

### Background

The frequent occurrence of public health emergencies poses a serious challenge to the emergency response capacity of Chinese nurses. However, nurses still suffer from insufficient emergency response knowledge and unskilled skills. An in-depth understanding of the risk factors affecting the emergency response capacity of nursing staff is needed to promote the overall improvement of the emergency response capacity of the healthcare system.

### Aims

To investigate the current situation and influencing factors of nursing staff's emergency response capacity for public health emergencies, and to provide a reference for the design of a standardized emergency response process, targeted training courses and revision of emergency plans.

### Methods

A descriptive cross-sectional survey was conducted from March to April 2024, using a self-designed general information questionnaire and a public health emergency response capacity scale for nursing staff. A survey was carried out among 4,680 nursing staff from secondary and above general hospitals in the Xinjiang region of China. Univariate analysis and multiple linear regression analysis were used to explore the influencing factors of nursing staff's emergency response capacity for public health emergencies.

**Data availability statement:** All relevant data are within the manuscript and its Supporting information files.

**Funding:** This study was supported by the 2022 research project of the Chinese Nursing Association (grant number ZHKY202212 to LZ) , the Tianshan Talents Medical and Health high-level Talents Training Program (grant number TSYC202301A041 to LZ), and the Excellent Talents and Innovative Team Training Program of the First Affiliated Hospital of Xinjiang Medical University (grant number cxtd202414 to LZ).

**Competing interests:** The authors have declared that no competing interests exist.

## Results

A total of 4292 valid questionnaires were collected. The average score on the emergency response capacity scale was 160.42±24.22, with a mean item score of 4.01±0.61. Multiple linear regression showed that sex, work experience, job title, position, rescue experience, emergency drill experience, presence of anxiety or fear, and willingness to rescue were the factors influencing the emergency response capacity of nursing staff for public health emergencies.

## Conclusions

Nursing staff in Xinjiang have the ability to deal with public health emergencies, but they still need to accumulate practical experience. Hospital administrators should improve the emergency response ability of nursing staff to address public health emergencies through a tiered training and assessment mechanism.

## Introduction

Since the early 21st century, there have been many public health emergencies in the world, including the H1N1 influenza pandemic, Ebola virus, Zika virus and COVID-19 [1–4], such emergencies have increased globally, marked by diverse causes, geographic variability, and complex hazards. China reported 36 natural disasters in 2015, making it the country with the highest number of natural disasters, and improving China's ability to respond to disaster events is critical to ensuring public safety [5]. As a crucial element of the rescue team for public health emergencies, nursing staff play an essential role throughout all stages of emergency management [6]. Once public health emergencies occur, nursing staff are required to undertake rescue operations promptly and orderly at the earliest time to minimize casualties and economic losses [7]. Emergency response capability refers to the ability of nursing staff to observe changes in patients' conditions in a timely and sensitive manner, make quick and correct judgments, and assist other medical staff in calmly and decisively taking emergency response measures in emergencies [8]. Studies have discovered that the emergency response ability of nurses directly determines the quality and outcome of patients' rescue [9,10]. Comprehensively understanding the emergency response status and ability level of nurses is an important approach to effectively control the development of public health emergencies [11]. Nevertheless, in contrast to other developed countries, the nursing research on public health emergencies and disasters in China is still in the exploratory stage, and a complete scientific system has not yet been established. The disaster relief capabilities and standards of nurses in China have not been unified yet, which is also the focus of our research [12].Therefore, an in-depth study on the emergency response capability of nursing staff for public health emergencies is crucial for ensuring patient safety and improving rescue efficiency and treatment. This study aimed to investigate the current status and influencing factors of the emergency response capacity of nursing staff in public health emergencies

in Xinjiang, in order to provide reference for promoting the development of the entire medical emergency system and the construction and improvement of emergency treatment base.

## Methods

### Study design

This cross-sectional study used convenience sampling to survey clinical nurses from grade II and above general hospitals in Xinjiang from March to April 2024. The convenience sampling method can quickly obtain large samples of data, avoid data lags due to complex sampling processes, and improve nurses' willingness to participate and questionnaire completion rates.

### Participants

Inclusion criteria: ① Nurses with a nursing license and registration; ② Nurses with at least 2 years of clinical experience; ③ Nurses who provided informed consent and voluntarily participated in the survey. Exclusion criteria: ① Nurses receiving further training, interns, and those working on probation; ② Nurses who were not on duty during the survey period for any reason. The sample size was calculated using the Kendall method, which required a sample size of more than 15 times the number of independent variables [13]. Accounting for a 20% invalid response rate, 4292 nurses were finally included. All participants in this study signed an informed consent form and was approved by the Institutional Review Board of the First Affiliated Hospital of Xinjiang Medical University (20220913-02).

### Survey tools

**General information questionnaire.** The investigators independently designed a 15-item questionnaire covering sex, age, education, title, position, and other relevant details. Among them, the titles are classified according to the professional title level of Chinese nurses as junior nurse, senior nurse, chief nurse, vice nurse consultant, and chief nurse practitioner. Positions are categorized by clinical nursing positions as nurse (none), group leader, head nurse, department head nurse, and director of nursing department.

### Emergency response capability scale for nurses in public health emergencies

As there is currently no unified standard for investigating the emergency response capabilities of nursing staff in public health emergencies in China, researchers and members of the research group adopted the PPRR theory [14] and the ICN Global Nurses' Disaster Nursing Competence Framework (2019) as guidance [15] through literature review and expert consultation. With reference to the Rating Index System of ICU Nurses' Emergency Response Ability for Major Infectious Diseases compiled by Wang [16], comprehensive management is implemented in four stages: pre-occurrence prevention, preparation, response during the outbreak, and recovery after the outbreak [17]. The scale preparation process is strictly followed. An assessment index system was initially formulated, and two rounds of Delphi expert consultation were conducted with experts in related fields. The contents were organized and revised based on expert feedback to construct the final assessment index system [18]. Compared with the shortcomings of existing scales that mostly focus on a single stage or lack theoretical integration[19,20], this indicator system is based on theoretical research, covers the whole cycle with multiple dimensions, and refines the specific competence requirements of nurses at each stage. In addition, the index system constructed in this study is applicable to all clinical nurses, which is more widely applicable compared with other studies [21,22]. Entries corresponding to the level 3rd indicators in the index system were converted to questionnaire entries, and a pre-survey was conducted among clinical nurses to examine the reliability and validity of the questionnaire. The Cronbach's α coefficient of the scale was 0.953, and the Cronbach's α coefficient of each dimension ranged from 0.932 to 0.974. The average content validity was 0.996, the KMO value of the questionnaire was 0.930, and

a total of 4 common factors were extracted through exploratory factor analysis. The cumulative variance contribution rate was 73.791%, signifying good reliability and validity. Based on the results, the entries were modified and optimized. The final scale included four dimensions, namely, the ability to prevent, prepare for, respond to, and recover from public health emergencies, with a total of 40 items. Each item was rated on a 5-point Likert scale from "very poorly" to "very well," with a total score range of 40–200 points. Higher scores indicated better emergency response capability. Criteria for evaluating the total emergency response capacity scale and the scores for each dimension: a score of 3 is used as a cut-off point, with an average score of less than 3 indicating a low capacity; a score of 3–4 indicating a medium capacity; and a score of 4 or more indicating a high capacity. The overall reliability of this scale in this study was 0.884 with Cronbach's α coefficient.

### Data collection

The online platform, WJX.cn, was used to distribute and recover the questionnaires. After obtaining consent from the hospital authorities, the QR code of the questionnaire was distributed to each department. The first page provided a brief overview of the purpose and significance of the survey, instructions for completing the questionnaire, precautions, and a statement ensuring voluntary participation and confidentiality. Each cell phone or computer allowed only one submission to prevent data duplication. After the survey was completed, the quality of the questionnaires was verified by two investigators, and invalid questionnaires with patterned responses or completion time < 2 min were excluded. Of the 4,380 questionnaires distributed, 4,292 valid questionnaires were recovered, resulting in a 97.99% effective recovery rate.

### Statistical analysis

Data entry and statistical analysis were performed by two investigators using SPSS 26.0. Measurement data were expressed as mean ± standard deviation ($\bar{x} \pm s$), and count data was described using frequency and percentage (%). One-way analysis was performed using a t-test and analysis of variance. Multiple linear regression analysis was performed with the emergency response capacity of nursing staff for public health emergencies as the dependent variable and the statistically significant variables in the univariate analysis as independent variables. Independent variable assignment includes: sex (male = 1, female = 2), years of working experience (with "≤ 5" as the reference group, dummy variables: ≤ 5 = 00000, 6–10 = 01000, 11–15 = 00100, 16–20 = 00010, and > 20 = 00001), job title (with "Junior nurse" as the reference group, dummy variables: junior nurse = 0000, senior nurse = 0100, chief nurse = 0010, and vice nurse consultant and above = 0001), position (with "none" as the reference group, dummy variables: none = 000, group leader = 010, head nurse and above = 001), rescue experience (none = 1, yes = 2), participation in emergency drills (no = 1, yes = 2), presence of anxiety or fear (no = 1, yes = 2), and willingness to rescue (unwilling = 1, willing = 2). Statistical significance was set at $P < 0.05$.

### Results

#### Score of emergency response capabilities of nurses for public health emergencies

Nurses scored an average of 160.42 ± 24.22 points on the emergency response capability scale for public health emergencies. The lowest scoring dimension was the ability to prevent public health emergencies (M = 3.87), followed by preparedness for and rescue during public health emergencies (M = 4.05). The highest score was for the ability to recover from public health emergencies (M = 4.08) (Fig 1).

#### Comparison of emergency response competency scores for public health emergencies among nurses with different demographic characteristics

A total of 4,292 nursing staff were surveyed in this study, with 257 males (6.0%) and 4,035 females (94.0%). The results of the univariate analysis showed statistically significant differences ($P < 0.05$) in emergency competence scores by sex,

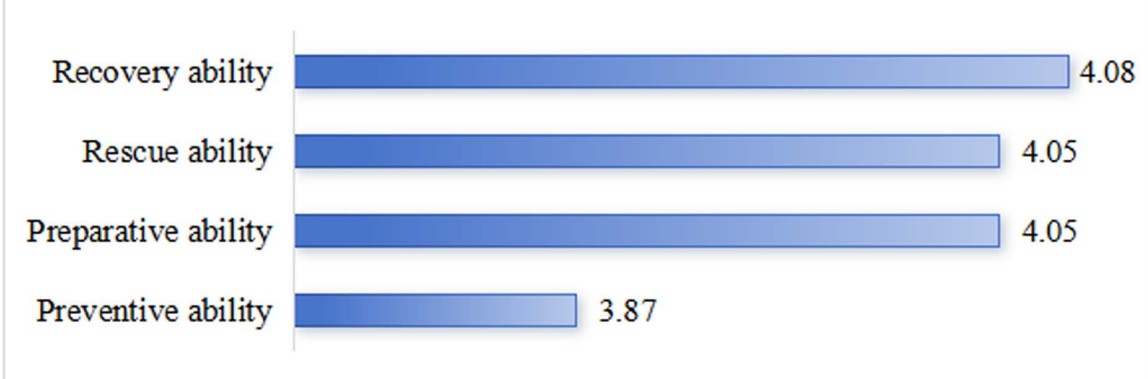

**Fig 1. Scores of nurses' emergency response capacity for public health emergencies.**

job title, position, department, years of working experience, rescue experience, whether emergency response training was received, participation in emergency drills, presence of anxiety or fear, and willingness to rescue (Table 1).

### Multiple linear regression analysis of the emergency response capacity of nursing staff for public health emergencies

The results of multiple linear regression showed that Sex, Years of work experience, Job title, Position, Rescue experience, Participation in emergency drills, Presence of anxiety or fear, and Willingness to rescue were the factors influencing the emergency response capacity of nursing staff for public health emergencies (Table 2).

## Discussion

### Analysis of the current situation of the emergency response capacity of nursing staff for public health emergencies

The results of this study showed that the total emergency response capacity score of nursing staff for public health emergencies was 160.42±24.22, with a mean item score of 4.01±0.61, which was high, slightly higher than the results of Taiwan [23], Korea [24], Iran [25], and the Philippines [26]. Analyzing the reasons, it may be because 55.9% of the nurses in this study were from tertiary hospitals, 89.5% of the nurses had received emergency response training, and 90.5% of the nurses had participated in drills, and the nurses had a high frequency of daily contact with patients with acute and critical illnesses, which improved their emergency judgment and disposal speed in repeated skills training and first aid practice, and effectively improved the nurses' ability to dispose of their patients in public health emergencies. In addition, Xinjiang, as an important region in the west, the Chinese government has continued to strengthen the construction of the public health system in recent years, and the establishment of a mobile disposal center for health emergencies in Xinjiang, which is able to effectively respond to and dispose of multiple types of public health emergencies and promote the establishment of standardized processes for emergency response by caregivers and the enhancement of the level of emergency response [27].

The lowest score among the four dimensions was for the ability to prevent public health emergencies (M = 3.87), which was consistent with the findings of previous studies [8]. The reason for this analysis may be that the early warning mechanism in Xinjiang is imperfect, and nurses are not clear about the workflow and division of responsibilities in the prevention and preparedness phases. In addition, the education in Xinjiang region is relatively backward, which leads to nurses' insufficient understanding of the importance of prevention, which in turn affects the quality of work. Hospitals can

**Table 1. Comparison of emergency response capacity scores for public health emergencies among nurses with different demographic characteristics.**

| Measurement item | Category | n (%) | (Mean±SD) | t/F | P |
|---|---|---|---|---|---|
| Sex | Male | 257(6.0) | 164.93±24.66 | 3.026 | 0.003 |
| | Female | 4035(94.0) | 160.13±24.17 | | |
| Age (years) | ≤25 | 482(11.2) | 158.53±24.74 | 2.417 | 0.065 |
| | 26~35 | 2228(51.9) | 160.61±23.51 | | |
| | 36~45 | 1237(28.8) | 161.41±24.81 | | |
| | >45 | 345(8.0) | 158.30±25.65 | | |
| Educational level | Secondary vocational school | 162 (3.8) | 159.54±27.42 | 2.320 | 0.073 |
| | College | 1550 (36.1) | 159.29±24.67 | | |
| | Bachelor's degree | 2568 (59.8) | 161.12±23.71 | | |
| | Master's degree or above | 12 (0.3) | 168.08±24.04 | | |
| Job title | Junior nurse | 1198 (27.9) | 158.97±24.64 | 4.891 | 0.002 |
| | Senior nurse | 1639 (38.2) | 162.18±23.75 | | |
| | Chief nurse | 1137 (26.5) | 159.50±24.39 | | |
| | Vice nurse consultant and above | 318 (7.4) | 160.09±23.95 | | |
| Position | None | 3394 (79.1) | 159.34±24.54 | 16.150 | <0.001 |
| | Group leader | 478 (11.1) | 164.60±23.36 | | |
| | Head nurse and above | 420 (9.8) | 164.36±21.58 | | |
| Department | General medicine | 1067 (24.9) | 162.43±24.31 | 4.299 | <0.001 |
| | Surgery | 1032 (24.0) | 157.93±24.88 | | |
| | Gynecology and obstetrics | 288 (6.7) | 163.00±22.86 | | |
| | Pediatrics | 186 (4.3) | 160.74±23.66 | | |
| | Acute and critical illness | 1054 (24.6) | 160.69±22.88 | | |
| | Operating rooms | 480 (11.2) | 158.30±25.71 | | |
| | Others | 185 (4.3) | 162.37±24.69 | | |
| Years of work experience (years) | ≤5 | 1058(24.7) | 158.72±23.95 | 3.389 | 0.009 |
| | 6~10 | 1072(25.0) | 160.51±23.46 | | |
| | 11~15 | 1108(25.8) | 161.46±24.45 | | |
| | 16~20 | 483(11.3) | 162.89±25.15 | | |
| | >20 | 571(13.3) | 159.29±24.67 | | |
| Only child | Yes | 766 (17.8) | 160.35±24.03 | 0.093 | 0.926 |
| | No | 3526 (82.2) | 160.44±24.27 | | |
| Marital status | Unmarried | 1121 (26.1) | 160.08±24.48 | 0.201 | 0.818 |
| | Married | 3047 (71.0) | 160.51±24.21 | | |
| | Others | 124 (2.9) | 161.20±22.23 | | |
| Hospital grading | Grade 2B | 94 (2.2) | 160.99±26.17 | 1.452 | 0.228 |
| | Grade 2A | 1685 (39.3) | 161.36±24.64 | | |
| | Grade 3B | 113 (2.6) | 159.95±26.45 | | |
| | Grade 3A | 2400 (55.9) | 159.76±23.72 | | |
| Rescue experience | Yes | 3285(76.5) | 162.32±23.70 | −9.155 | <0.001 |
| | No | 1007 (23.5) | 154.21±24.86 | | |
| Participation in emergency training | Yes | 3842(89.5) | 161.64±23.68 | −9.004 | <0.001 |
| | No | 450(10.5) | 150.00±26.20 | | |
| Participation in emergency drills | Yes | 3884(90.5) | 161.60±23.72 | −9.262 | <0.001 |
| | No | 408(9.5) | 149.16±26.03 | | |

*(Continued)*

**Table 1.** (Continued)

| Measurement item | Category | n (%) | (Mean±SD) | t/F | P |
|---|---|---|---|---|---|
| Presence of anxiety or fear | Yes | 1645(38.3) | 156.36±24.34 | 8.689 | <0.001 |
| | No | 2647(61.7) | 162.94±23.80 | | |
| Willingness to rescue | Yes | 3834(89.3) | 161.93±23.49 | −10.945 | <0.001 |
| | No | 458(10.7) | 147.76±26.50 | | |

**Table 2.** Multiple linear regression analysis of the factors influencing the emergency response capacity of nursing staff for public health emergencies.

| Variables | | B | SE | β | t | P | Tolerance | VIF |
|---|---|---|---|---|---|---|---|---|
| (Constant) | | 127.670 | 4.363 | – | 29.259 | <0.001 | – | – |
| Sex | | −3.623 | 1.541 | −0.035 | −2.351 | 0.019 | 0.936 | 1.068 |
| Years of work experience (years) | 6–10 | 0.384 | 1.226 | 0.007 | 0.314 | 0.754 | 0.445 | 2.249 |
| | 11–15 | 2.893 | 1.354 | 0.052 | 2.137 | 0.033 | 0.356 | 2.806 |
| | 16–20 | 5.715 | 1.656 | 0.075 | 3.450 | 0.001 | 0.457 | 2.189 |
| | >20 | 3.547 | 1.735 | 0.050 | 2.044 | 0.041 | 0.360 | 2.776 |
| Job title | Senior nurse | 0.650 | 1.143 | 0.013 | 0.569 | 0.570 | 0.406 | 2.463 |
| | Chief nurse | −4.826 | 1.414 | −0.088 | −3.413 | 0.001 | 0.321 | 3.111 |
| | Vice nurse consultant and above | −7.254 | 2.179 | −0.078 | −3.329 | 0.001 | 0.384 | 2.602 |
| Position | Group leader | 4.563 | 1.184 | 0.059 | 3.854 | <0.001 | 0.902 | 1.108 |
| | Head nurse and above | 6.011 | 1.442 | 0.074 | 4.170 | <0.001 | 0.682 | 1.466 |
| Rescue experience | | 5.386 | 0.898 | 0.094 | 5.998 | <0.001 | 0.864 | 1.157 |
| Participation in emergency drills | | 8.188 | 1.282 | 0.099 | 6.386 | <0.001 | 0.885 | 1.130 |
| Presence of anxiety or fear | | −6.349 | 0.732 | −0.127 | −8.669 | <0.001 | 0.987 | 1.013 |
| Willingness to rescue | | 11.603 | 1.165 | 0.148 | 9.958 | <0.001 | 0.967 | 1.034 |

Note: $R=0.296$, $R^2=0.087$, Adjusted $R^2=0.084$, $F=29.230$, $P<0.001$.

improve their emergency response ability by designing simulation scenarios of public health emergencies that are close to the actual situation in Xinjiang, so that nurses can familiarize themselves with the prevention and preparedness process during simulation exercises. Clarify the responsibilities of nurses in prevention work and establish a reasonable incentive mechanism to improve the work motivation of nurses. At the same time, strengthen the cooperation between nurses and the community, establish a community emergency response network, and improve the overall awareness and ability of prevention.

## Analysis of factors influencing nursing staff's ability to respond to public health emergencies

This study revealed that male nurses had a higher emergency response capacity for public health emergencies than female nurses, consistent with the findings of other scholars [28,29]. This may be related to the fact that most male nurses are concentrated in the emergency department, ICU or operating room, and they have rich clinical experience, are better than female nurses in patient handling and instrument and equipment operation, and are able to remain calm, quickly adjust their state and respond positively to emergencies. It was found that men's resilience was higher than women's, and the psychosomatic impact of frontline tasks on women was greater than that of men, suggesting that men may be chosen more often in the course of performing critical tasks when professional needs are met [30]. Statistically significant differences in the emergency response ability scores of nursing staff with different years of working experience were

noted, which was consistent with the findings of Luo et al [31]. This study showed that nurses with 11 years of working experience and above had higher emergency response capacity compared to nurses with ≤5 years of working experience, likely because of their longer experience in clinical practice and better ability to identify and address patient needs, leading to higher-quality care. Moreover, experienced nurses are generally more skilled in handling public health emergencies, with significantly higher response scores than less experienced nurses [32]. For nurses with shorter working years and less experience in emergency response, experts can be invited to organize theoretical lectures on emergency response to public health emergencies to help them quickly establish an emergency response knowledge system. For nurses with rich working experience, hospital administrators can arrange for them to participate in advanced emergency management training courses to learn advanced emergency management concepts and methods at home and abroad, and improve their emergency management capabilities.While it is generally believed that nurses with higher titles are more competent in emergency situations [8], this study showed that nurses with medium and senior titles had lower emergency response capacity scores than those with junior titles, with no intergroup differences between junior and senior nurses. This may be because nurses with higher titles are more focused on management, teaching, or scientific research and have been away from clinical nursing for a long time, leading to a decline in their first-aid knowledge and skills [11]. In addition, since disaster nursing education is still developing and lacks a well-established educational discipline system [33], most senior nurses have not been exposed to basic disaster nursing courses, further reducing their emergency response capacities [34]. Emergency response capacity varied by position. Nurses who were group leaders or head nurses and above had higher emergency response capacities than those without any leadership roles. This could be attributed to their professionalism and leadership skills, which are crucial in emergency situations [35]. These nurses can make sound decisions, manage nursing issues, and effectively organize and lead teams in patient rescue efforts. Managers can provide more professional development opportunities for nursing staff who do not hold positions, such as attending domestic and international emergency management seminars and academic exchange activities, to enhance their professionalism and emergency response capabilities.

The results of multiple linear regression showed that nurses with rescue experience, who had participated in emergency drills, who were not anxious or fearful of emergencies, and who were willing to perform rescue operations had higher emergency response capacity. This may be because nurses with rescue experience gain practical experience from participating in actual emergency rescue operations. Moreover, they had undergone multiple trainings and assessments in first aid courses [36], consolidating their emergency response capacity through practical operations [37,24]. Hospital administrators can organize nursing staff with rescue experience to carry out experience sharing sessions, so that they can share their personal experience and valuable experience in the rescue process, providing reference for other nursing staff. For nursing staff without rescue experience, simulated practical exercises can be carried out, allowing them to carry out emergency response in simulated scenarios to enhance their emergency response practical ability. Emergency drills provide a simulated environment where nurses can fully engage in rescue operations and receive timely corrections, preventing any real harm to the patients. Participation in emergency drills enhances communication skills and teamwork, further improving the emergency response capacity and teamwork among nurses [38]. In this study, 38.3% of nurses experienced anxiety or fear before participating in emergency rescues, while 89.3% were willing to participate in the rescue. This anxiety may stem from concerns about isolation from family and society, the risk of infection [39,40], and doubts about their ability to handle critical situations, all of which could increase psychological stress and lower emergency response capacity. It is recommended that hospital administrators establish a psychological counseling service mechanism to provide timely psychological support to nursing staff to help them relieve work stress and anxiety. Previous studies have shown that Chinese nurses have a high willingness to respond to public health emergencies [41] and demonstrate strong emergency response capacity. Nurses have a high sense of responsibility [42] as they are willing to prioritize patient care, even at personal risk. They are dedicated to saving lives and fully commit to their duties during rescues. Hospitals should encourage nurses to participate in the rescue work of public health emergencies and

cultivate their enthusiasm and initiative so that they can give full play to their strengths and abilities and improve the success rate of patients' treatment.

Xinjiang region is characterized by remote geographical location, lagging economic development, and ethnic and cultural diversity, and the spatial allocation of healthcare resources exhibits regional inequity [43]. The findings are highly applicable in regions with similar healthcare structures or cultural backgrounds to Xinjiang, such as Tibet and Inner Mongolia in China. In the future, pilot applications can be prioritized in these regions, and ultimately a nursing staff emergency response capacity evaluation tool with both regional characteristics and strong scientific generality can be formed.

## Limitations

This study had several limitations. First, the cross-sectional design only observed the variables at a specific time point, preventing the tracking of individual trends over time, and thus, causality cannot be confirmed. In the future, longitudinal studies can be applied to conduct joint investigations in multiple regions to test the causal relationship of this research. Second, the questionnaire was self-filled by the nursing staff online, which is subjective and the results of the survey may not reflect the true values. In the future, the measurement of objective indicators can be introduced to strengthen the methodological rigor, thus greatly improving the generalizability and explanatory power of the study. Third, the convenience sampling method used in this study only surveyed subjects from one province in China, which lacked representativeness and limited the generalizability and extrapolation of the findings. In the future, multicenter studies can be conducted nationwide to expand the sample size and test the reliability of the findings.

## Conclusion

The emergency response capacity of nursing staff for public health emergencies in Xinjiang was high. Higher emergency response capacities were observed in male nurses, nurses with 11 or more years of working experience, junior titles, specific positions, rescue experience, participation in emergency drills, and those who were not anxious or fearful and were willing to rescue. Hospital administrators can carry out a tiered and categorized training system based on nurses with different emergency response levels, such as basic skills reinforcement and higher-order competency expansion. Dynamically adjust the training focus by conducting multi-sectoral joint simulation exercises. Establish a policy guarantee and incentive mechanism to directly link participation in rescue and training outcomes with career development, and enhance nurses' ability to respond to public health emergencies through a series of measures.

## Supporting information

**S1 File. Sample questionnaire.**
(XLSX)

**S2 Fig. Scores of nurses' emergency response capacity for public health emergencies.**
(TIF)

## Author contributions

**Conceptualization:** Ping Yan, Xia Zhang.

**Formal analysis:** Hu Peng, Zhijie Cao.

**Investigation:** Ping Yan, Xia Zhang.

**Methodology:** Hu Peng, Zhijie Cao.

**Supervision:** Xin Gu, Lina Ma.

**Validation:** Ping Yan, Lina Ma.

**Writing – original draft:** Xin Gu, Li Zhang.

**Writing – review & editing:** Xin Gu, Li Zhang.

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
