## [Decision Letter · Decision Letter 0]

11 Feb 2025

PONE-D-24-51537Analysis of the emergency response capacity of nursing staff for public health emergencies and the influencing factors: A cross-sectional study in ChinaPLOS ONE

Dear Dr.  Zhang,

Thank you for submitting your manuscript to PLOS ONE. After careful consideration, we feel that it has merit but does not fully meet PLOS ONE’s publication criteria as it currently stands. Therefore, we invite you to submit a revised version of the manuscript that addresses the points raised during the review process.

We look forward to receiving your revised manuscript.

Kind regards,

Mohamed Gamal Elsehrawy

Academic Editor

PLOS ONE

Journal Requirements:

“This work were supported by the 2022 research project of the Chinese Nursing Association grant number ZHKY202212、Xinjiang Key Laboratory of Medical Animal Mode Research and Excellent Talents and Innovative Team Training Program of the First Affiliated Hospital of Xinjiang Medical University grant number cxtd202414.”

5. We note that your Data Availability Statement is currently as follows: “All relevant data are within the manuscript and in Supporting Information files.”

7. Please remove your figures from within your manuscript file, leaving only the individual TIFF/EPS image files, uploaded separately. These will be automatically included in the reviewers’ PDF.

Reviewers' comments:

Reviewer's Responses to Questions

**Comments to the Author**

1. Is the manuscript technically sound, and do the data support the conclusions?

Reviewer #1: Yes

Reviewer #2: Yes

2. Has the statistical analysis been performed appropriately and rigorously? 

Reviewer #1: Yes

Reviewer #2: Yes

3. Have the authors made all data underlying the findings in their manuscript fully available?

Reviewer #1: Yes

Reviewer #2: Yes

4. Is the manuscript presented in an intelligible fashion and written in standard English?

Reviewer #1: Yes

Reviewer #2: Yes

5. Review Comments to the Author

Reviewer #1: 1. Clarity of Objectives:

While the study's objectives are clear, emphasizing how the findings directly inform the design and implementation of emergency response programs for nurses would enhance their practical relevance. Consider connecting the results to specific programmatic improvements.

2. Sampling Method:

The rationale for using convenience sampling should be elaborated, especially given the large sample size. Discuss potential selection biases and how they might influence the study's results and conclusions.

3. Instrument Validation:

The validation process for the self-designed emergency response capability scale is commendable. However, clarify why this tool was preferred over established, validated scales. Address its unique advantages or alignment with the study’s context.

4. Data Presentation:

While the results are comprehensive, presenting key findings visually (e.g., bar charts, pie charts, or scatterplots) could enhance readability and facilitate better understanding of the data.

5. Influencing Factors:

The identification of significant factors through multiple linear regression is valuable. Discuss how these factors can be operationalized into actionable training programs or policies, providing concrete examples where possible.

6. Generalizability:

The study's findings are specific to Xinjiang. Discuss their applicability to regions with similar healthcare structures or cultural contexts, and suggest areas for further comparative research to test their broader relevance.

7. Discussion Depth:

Gaps in prevention and preparedness scores should be further explored. Suggest targeted interventions such as simulation-based training, policy updates, or community engagement strategies to address these weaknesses.

8. Ethical Considerations:

Ethical approval and participant consent are noted, but provide more detail on the steps taken to ensure confidentiality and protect sensitive information, especially during data collection and analysis.

9. Limitations:

The limitations section should explicitly address the reliance on self-reported data and the risk of response bias. Discuss how these factors might have influenced the findings and what measures were taken to mitigate them.

10. Implications:

Highlight actionable recommendations for nursing administrators and policymakers. For example, propose targeted training modules, stress management programs, and leadership development initiatives to strengthen emergency response capacity.

Reviewer #2: Analysis of the emergency response capacity of nursing staff for public health

emergencies and the influencing factors: A cross-sectional study in China

Xin Gu, Ping Yan, Xia Zhang, Hu Peng, Zhijie Cao, Lina Ma, Li Zhang

This manuscript presents a cross-sectional study that aims to investigate the public health emergency response capacity of nursing staff and relevant influencing factors in order to inform preparedness programs against future emergencies.

Researchers constructed and evaluated an assessment tool that comprised mainly of Likert-scale questions investigating four categories of nurses’ ability to prevent, prepare, rescue and recover from public health emergencies they needed to cope with.

They used convenience sampling to collect data from 4292 nurses whose average score on emergency response capacity scale was 160.42 ± 24.22, with a mean score per item of 4.01 ± 0.61. Multiple linear regression was performed and showed that higher emergency response capacities were observed in male nurses, nurses with 11 or more years of working experience, junior titles, specific positions, rescue experience, participation in emergency drills, and those who were not anxious or fearful and were willing to rescue.

Existing literature on the subject is available for many areas in the world including China, but it deserves further study as public health emergencies seem to be issues that will concern us in the near future. The authors concluded that the emergency response capacity of nursing staff for public health emergencies in Xinjiang was high and that hospital administrators should develop intervention programs targeting the specific areas of weakness that arose from the study and tailored to different levels of emergency response capacity to strengthen education and practical drills for nursing staff.

The study’s main strength is the large sample and the rigorous methodology of the study design. On the other hand, the limitations are apparent as the study was performed in only one region which prevents the generalizability of results, while convenience sampling prevents representativeness even for this area. Also, the statistical approach used requires assumptions that are not easily perceived.

The manuscript adheres to PLOS ONE guidelines for authors and mostly to the STROBE checklist and SAMPL guidelines. In my opinion, the manuscript in general appropriately discusses the claims made and in terms of technical completeness, with an adequate statistical analysis for the available data and can be published after revisions are made.

More details and revision suggestions are presented below, separately for each section.

Abstract and introduction

The abstract is concise and informative, but the background section should include a phrase on the current situation of nurses’ emergency response capacity and not the last sentence which is actually a recommendation.

The introduction section is useful and sets the background for the study. The authors generally identify literature on the topic and how the study relates to it. However, the first paragraph that depicts the global situation regarding public health emergencies is documented with references that are out of scope and their conclusions or results are not about the arguments made in this section (references 1-7).

Methods

While the whole process of designing, assessing and scoring the evaluation tools used is explicitly described, this excerpt “Criteria for emergency ability: the average score of items < 3 points indicates low emergency ability; 3 to 4 points indicates medium emergency ability; > 4 points indicates high emergency ability.” is not well understood and since figure 1 is based on it, it would be better to elaborate. Also, references 16,17 are out of scope and while they refer to the same point, they do not document it.

Finally, the description of dummy variables used in multiple linear regression belongs in the methodology section instead of the results.

Results

The authors choose to use an assessment tool with Likert scales and then translating the scores as a continuous variable and use multiple linear regression in the analysis. While this seems a technically legitimate way to manipulate the results, it makes understanding them rather difficult for the reader because Likert scales lead to ordinal perception of the results. Therefore, I would prefer describing the results, and especially the direction they should be interpreted, in more detail. Furthermore presenting the four categories (preventative, preparation, rescue and recovery ability) with low, medium and high scoring confuses readers’ understanding.

Also nurse rank, job title and position mentioned both in the results and the tables are not familiar to international audiences, which is the target here, and need to be explained.

Discussion

The discussion overall supports the results and uses relatable references to document the arguments, although some exceptions exist. In the first paragraph the clip “During the COVID-19 pandemic, the remarkable performance of Chinese nurses worldwide not only demonstrates the advancement of China's medical and health undertakings but also showcases China's contribution to global public health” not only is not backed by the bibliography as it is not referenced, it also seems irrelevant to the study’s results.

In the second paragraph nurses’ low score in the ability to prevent public health emergencies could be explained mainly by the study’s design which collects data from secondary and tertiary care hospitals, where nurses are not expected to perform health prevention services (at least to my knowledge which does not include the chines health system).

My main disagreement though, is the issue of male-to-female comparison described in the third paragraph. First of all, the fact that only 6% of the respondents are male should make the authors cautious about discussing this result in a conclusive way as the very small male-to-female ratio yields statistical doubts. Then, the justification presented should also be documented with strong arguments and references as it is rather sexist since the authors discuss that “male nurses’ physical strength, rational thinking, and decisive actions enable them to handle heavy workloads and adapt to challenging environments” and “have higher adaptability and resistance to stress, allowing them to remain calm and quickly adjust to sudden, complex situations”. On that note, reference 27 studies only male nurses and cannot build an argument on male-to-female comparison and I cannot find reference 28 to form an opinion.

Limitations

The authors successfully describe the lack of generalizability of their results and the difficulty of addressing causality. Other issues that have to be addressed though, are the method of sampling (convenience sampling) which further limits representativeness, and coping with bias which is not mentioned.

Figures and tables

The tables are complete and self-explanatory. The cruciform grid might benefit from drawing the axes to help the reader better assign the values mentioned in their minds.

References

Some referencing mistakes need to be corrected, e.g.6 and other related issues are reported within the sections referenced.

6. PLOS authors have the option to publish the peer review history of their article (what does this mean? ). If published, this will include your full peer review and any attached files.

**Do you want your identity to be public for this peer review?** For information about this choice, including consent withdrawal, please see our Privacy Policy .

Reviewer #1: **Yes: ** Ateya Megahed Ibrahim

Reviewer #2: No

---

## [Author Response · Author response to Decision Letter 1]

28 Mar 2025

Dear Editor:

Thank you for giving us the opportunity to submit a revised draft of the manuscript “Analysis of the emergency response capacity of nursing staff for public health emergencies and the influencing factors: A cross-sectional study in China” (PONE-D-24-51537). We sincerely appreciate the time and effort that you and the reviewers have dedicated to providing feedback on our manuscript. We are grateful for the valuable insights, which has considerably improved our paper. In this revised version, changes to our manuscript were all highlighted in red. The following is our response to your proposed changes.

We hope that the changes we’ve made resolve all your concerns about the article. We are more than happy to make any further changes that will improve the paper and facilitate successful publication.

Sincerely,

Li Zhang

March 28, 2025

Response to the comments of Reviewer #1

Q1. Clarity of Objectives:

While the study's objectives are clear, emphasizing how the findings directly inform the design and implementation of emergency response programs for nurses would enhance their practical relevance. Consider connecting the results to specific programmatic improvements.

Response: Thank you very much for your suggestion, and the authors have revised and added to the aims section: To investigate the current situation and factors affecting the emergency response capacity of nursing staff for public health emergencies, and to provide a reference for the design of a standardized emergency response process, a targeted training course, and a revision of the emergency response plan.

Q2. Sampling Method:

The rationale for using convenience sampling should be elaborated, especially given the large sample size. Discuss potential selection biases and how they might influence the study's results and conclusions.

Response: Thank you for the valuable comment. We have added to the Study design section: The convenience sampling method can quickly obtain large samples of data, avoid data lags due to complex sampling processes, and improve nurses' willingness to participate and questionnaire completion rates. The authors have provided an additional explanation of the limitations of convenience sampling in the Restrictions section.

Q3. Instrument Validation:

The validation process for the self-designed emergency response capability scale is commendable. However, clarify why this tool was preferred over established, validated scales. Address its unique advantages or alignment with the study’s context.

Response: Thank you for your valuable suggestions, the author has made additional clarifications, which include the following: Compared with the existing scales which mostly focus on a single stage or lack of theoretical integration [19, 20], this indicator system is based on theoretical research, covers the whole cycle with multi-dimensions, and refines the specific competence requirements of nurses in each stage. In addition, the index system constructed in this study is applicable to all clinical nurses, which is more widely applicable compared with other studies [21, 22].

Q4. Data Presentation:

While the results are comprehensive, presenting key findings visually (e.g., bar charts, pie charts, or scatterplots) could enhance readability and facilitate better understanding of the data.

Response: Thank you very much for your suggestion, the authors have modified Figure 1 to a bar chart that facilitates better understanding by the reader.

Q5. Influencing Factors:

The identification of significant factors through multiple linear regression is valuable. Discuss how these factors can be operationalized into actionable training programs or policies, providing concrete examples where possible.

Response: Thank you very much for your expert advice, the authors have made the following revisions in the Discussion section: For nurses with shorter working years and less experience in emergency response, experts can be invited to organize theoretical lectures on emergency response to public health emergencies to help them quickly establish an emergency response knowledge system. For nurses with rich working experience, hospital administrators can arrange for them to participate in advanced emergency management training courses to learn advanced emergency management concepts and methods at home and abroad, and improve their emergency management capabilities. Managers can provide more professional development opportunities for nursing staff who do not hold positions, such as attending domestic and international emergency management seminars and academic exchange activities, to enhance their professionalism and emergency response capabilities. Hospital administrators can organize nursing staff with rescue experience to carry out experience sharing sessions, so that they can share their personal experience and valuable experience in the rescue process, providing reference for other nursing staff. For nursing staff without rescue experience, simulated practical exercises can be carried out, allowing them to carry out emergency response in simulated scenarios to enhance their emergency response practical ability. It is recommended that hospital administrators establish a psychological counseling service mechanism to provide timely psychological support to nursing staff to help them relieve work stress and anxiety. Hospitals should encourage nurses to participate in the rescue work of public health emergencies and cultivate their enthusiasm and initiative so that they can give full play to their strengths and abilities and improve the success rate of patients' treatment.

Q6. Generalizability:

The study's findings are specific to Xinjiang. Discuss their applicability to regions with similar healthcare structures or cultural contexts, and suggest areas for further comparative research to test their broader relevance.

Response: Thank you for your expert guidance, which the authors have added to the discussion section. The details are as follows: Xinjiang region is characterized by remote geographical location, lagging economic development, and ethnic and cultural diversity, and the spatial allocation of healthcare resources exhibits regional inequity [44]. The findings are highly applicable in regions with similar healthcare structures or cultural backgrounds to Xinjiang, such as Tibet and Inner Mongolia in China. In the future, pilot applications can be prioritized in these regions, and ultimately a nursing staff emergency response capacity evaluation tool with both regional characteristics and strong scientific generality can be formed.

Q7. Discussion Depth:

Gaps in prevention and preparedness scores should be further explored. Suggest targeted interventions such as simulation-based training, policy updates, or community engagement strategies to address these weaknesses.

Response: Thank you for your suggestion, which the author has added to the discussion section. The details are as follows The reason for this analysis may be that the early warning mechanism in Xinjiang is imperfect, and nurses are not clear about the workflow and division of responsibilities in the prevention and preparedness phases. In addition, the education in Xinjiang region is relatively backward, which leads to nurses' insufficient understanding of the importance of prevention, which in turn affects the quality of work. Hospitals can improve their emergency response ability by designing simulation scenarios of public health emergencies that are close to the actual situation in Xinjiang, so that nurses can familiarize themselves with the prevention and preparedness process during simulation exercises. Clarify the responsibilities of nurses in prevention work and establish a reasonable incentive mechanism to improve the work motivation of nurses. At the same time, strengthen the cooperation between nurses and the community, establish a community emergency response network, and improve the overall awareness and ability of prevention.

Q8. Ethical Considerations:

Ethical approval and participant consent are noted, but provide more detail on the steps taken to ensure confidentiality and protect sensitive information, especially during data collection and analysis.

Response: Thank you for the special reminder that ethical considerations were not mentioned at the end according to the journal's typesetting requirements, so the authors have removed the ethical section in order to protect sensitive information.

Q9. Limitations:

The limitations section should explicitly address the reliance on self-reported data and the risk of response bias. Discuss how these factors might have influenced the findings and what measures were taken to mitigate them.

Response: Thank you for your professional advice, the authors have made corrections and additional clarifications in the limitations section.

Q10. Implications:

Highlight actionable recommendations for nursing administrators and policymakers. For example, propose targeted training modules, stress management programs, and leadership development initiatives to strengthen emergency response capacity.

Response: Thank you for your suggestion, the authors have revised the description in the conclusion section as follows: Hospital administrators can carry out a tiered and categorized training system based on nurses with different emergency response levels, such as basic skills reinforcement and higher-order competency expansion. Dynamically adjust the training focus by conducting multi-sectoral joint simulation exercises. Establish a policy guarantee and incentive mechanism to directly link participation in rescue and training outcomes with career development, and enhance nurses' ability to respond to public health emergencies through a series of measures.

Response to the comments of Reviewer #2

Q1. Abstract and introduction

The abstract is concise and informative, but the background section should include a phrase on the current situation of nurses’ emergency response capacity and not the last sentence which is actually a recommendation.

Response: Thank you very much for your valuable suggestions, we have reworked the background section in the manuscript with the following changes: The frequent occurrence of public health emergencies poses a serious challenge to the emergency response capacity of Chinese nurses. However, nurses still suffer from insufficient emergency response knowledge and unskilled skills. An in-depth understanding of the risk factors affecting the emergency response capacity of nursing staff is needed to promote the overall improvement of the emergency response capacity of the healthcare system.

The introduction section is useful and sets the background for the study. The authors generally identify literature on the topic and how the study relates to it. However, the first paragraph that depicts the global situation regarding public health emergencies is documented with references that are out of scope and their conclusions or results are not about the arguments made in this section (references 1-7).

Response: Thanks to your suggestions, the authors have reorganized and rewritten the introduction section, and the references have been cited in a way that is more relevant to the topic of the study and makes the whole text more logical.

Q2. Methods

While the whole process of designing, assessing and scoring the evaluation tools used is explicitly described, this excerpt “Criteria for emergency ability: the average score of items < 3 points indicates low emergency ability; 3 to 4 points indicates medium emergency ability; > 4 points indicates high emergency ability.” is not well understood and since figure 1 is based on it, it would be better to elaborate. Also, references 16,17 are out of scope and while they refer to the same point, they do not document it.

Response: Thank you for your suggestions, the authors have made the appropriate changes as follows: Criteria for evaluating the total emergency response capacity scale and the scores for each dimension: a score of 3 is used as a cut-off point, with an average score of less than 3 indicating a low capacity; a score of 3 to 4 indicating a medium capacity; and a score of 4 or more indicating a high capacity.

Finally, the description of dummy variables used in multiple linear regression belongs in the methodology section instead of the results.

Response: Thank you for your careful scrutiny. We apologize for our carelessness. Based on your comments, we have written a description of the dummy variables used in multiple linear regression in the methodology section.

Q3. Results:

The authors choose to use an assessment tool with Likert scales and then translating the scores as a continuous variable and use multiple linear regression in the analysis. While this seems a technically legitimate way to manipulate the results, it makes understanding them rather difficult for the reader because Likert scales lead to ordinal perception of the results. Therefore, I would prefer describing the results, and especially the direction they should be interpreted, in more detail. Furthermore presenting the four categories (preventative, preparation, rescue and recovery ability) with low, medium and high scoring confuses readers’ understanding.

Response: The criticisms you have raised are very pertinent and relevant. The authors have visualized the results of the Prevention, Preparedness, Rescue and Resilience scores in a bar chart for better understanding by the reader, see Figure 1.

Also nurse rank, job title and position mentioned both in the results and the tables are not familiar to international audiences, which is the target here, and need to be explained.

Response: Thank you for your suggestion, the authors have provided additional clarification in the Methods section as follows: Among them, the titles are classified according to the professional title level of Chinese nurses as junior nurse, senior nurse, chief nurse, vice nurse consultant, and chief nurse practitioner. Positions are categorized by clinical nursing positions as nurse (none), group leader, head nurse, department head nurse, and director of nursing department.

Q4. Discussion

The discussion overall supports the results and uses relatable references to document the arguments, although some exceptions exist. In the first paragraph the clip “During the COVID-19 pandemic, the remarkable performance of Chinese nurses worldwide not only demonstrates the advancement of China's medical and health undertakings but also showcases China's contribution to global public health” not only is not backed by the bibliography as it is not referenced, it also seems irrelevant to the study’s results.

Response: Thank you for your criticism and correction, the authors have reworked this section and cited literature to support it.

In the second paragraph nurses’ low score in the ability to prevent public health emergencies could be explained mainly by the study’s design which collects data from secondary and tertiary care hospitals, where nurses are not expected to perform health prevention services (at least to my knowledge which does not include the chines health system).

Response: Thank you for your suggestion, the author has revised this section.

My main disagreement though, is the issue of male-to-female comparison described in the third paragraph. First of all, the fact that only 6% of the respondents are male should make the authors cautious about discussing this result in a conclusive way as the very small male-to-female ratio yields statistical doubts. Then, the justification presented should also be documented with strong arguments and references as it is rather sexist since the authors discuss that “male nurses’ physical strength, rational thinking, and decisive actions enable them to handle heavy workloads and adapt to challenging environments” and “have higher adaptability and resistance to stress, allowing them to remain calm and quickly adjust to sudden, complex situations”. On that note, reference 27 studies only male nurses and cannot build an argument on male-to-female comparison and I cannot find reference 28 to form an opinion.

Response: Thank you very much for your valuable sugges

---

## [Decision Letter · Decision Letter 1]

21 Apr 2025

Analysis of the emergency response capacity of nursing staff for public health emergencies and the influencing factors: A cross-sectional study in China

PONE-D-24-51537R1

Dear Author,

We’re pleased to inform you that your manuscript has been judged scientifically suitable for publication and will be formally accepted for publication once it meets all outstanding technical requirements.

Kind regards,

Mohamed Gamal Elsehrawy

Academic Editor

PLOS ONE

Additional Editor Comments (optional):

Reviewers' comments:

Reviewer's Responses to Questions

**Comments to the Author**

1. If the authors have adequately addressed your comments raised in a previous round of review and you feel that this manuscript is now acceptable for publication, you may indicate that here to bypass the “Comments to the Author” section, enter your conflict of interest statement in the “Confidential to Editor” section, and submit your "Accept" recommendation.

Reviewer #1: All comments have been addressed

Reviewer #2: All comments have been addressed

2. Is the manuscript technically sound, and do the data support the conclusions?

Reviewer #1: Yes

Reviewer #2: Yes

3. Has the statistical analysis been performed appropriately and rigorously? 

Reviewer #1: Yes

Reviewer #2: Yes

4. Have the authors made all data underlying the findings in their manuscript fully available?

Reviewer #1: Yes

Reviewer #2: Yes

5. Is the manuscript presented in an intelligible fashion and written in standard English?

Reviewer #1: No

Reviewer #2: Yes

6. Review Comments to the Author

Reviewer #1: Thanks alot for your great efforts in addressing all the required comments and congratulations and hope you all success

Reviewer #2: (No Response)

7. PLOS authors have the option to publish the peer review history of their article (what does this mean? ). If published, this will include your full peer review and any attached files.

**Do you want your identity to be public for this peer review?** For information about this choice, including consent withdrawal, please see our Privacy Policy .

Reviewer #1: No

Reviewer #2: No

---

## [Editor Report · Acceptance letter]

PONE-D-24-51537R1

PLOS ONE

Dear Dr. Zhang,

I'm pleased to inform you that your manuscript has been deemed suitable for publication in PLOS ONE. Congratulations! Your manuscript is now being handed over to our production team.

Kind regards,

on behalf of

Dr. Mohamed Gamal Elsehrawy

Academic Editor

PLOS ONE